# Suicide Interventions in Spain and Japan: A Comparative Systematic Review

**DOI:** 10.3390/healthcare12070792

**Published:** 2024-04-06

**Authors:** Noelia Lucía Martínez-Rives, María del Pilar Martín Chaparro, Bibha Dhungel, Stuart Gilmour, Rory D. Colman, Yasuhiro Kotera

**Affiliations:** 1Department of Psychiatry and Social Psychology, University of Murcia, 30100 Murcia, Spain; noelialucia.martinezr@um.es; 2School of International Liberal Studies, Waseda University, Tokyo 169-0051, Japan; bibha-dhungel@umin.ac.jp; 3Department of Health Policy, National Centre for Child Health and Development, Tokyo 157-0074, Japan; 4Graduate School of Public Health, St. Luke’s International University, Tokyo 104-0045, Japan; sgilmour@slcn.ac.jp; 5College of Health, Psychology and Social Care, University of Derby, Derby DE22 1GB, UK; rory.d.colman@gmail.com; 6Faculty of Medicine and Health Sciences, University of Nottingham, Nottingham NG7 2TU, UK; yasuhiro.kotera@nottingham.ac.uk; 7Center for Infectious Disease Education and Research, Osaka University, Suita 565-0871, Japan

**Keywords:** suicide, psychological intervention, Spain, Japan, systematic review

## Abstract

(1) Background: This systematic review presents an overview of psychological interventions in suicide published between 2013 and 2023 in Spain and Japan, sparked by Spain’s alarming recent increase in suicide rates and the potential exemplar of Japan’s reduction efforts. (2) Methods: Following the PRISMA checklist, the databases Web of Science, Scopus, PubMed, and PsycInfo were searched using the terms [(“suicide” OR “suicidal behavior” OR “suicidal attempt” OR “suicidal thought” OR “suicidal intention”) AND (“prevention” OR “intervention” OR “psychosocial treatment” OR “Dialectical Behavior Therapy” OR “Cognitive Therapy” OR “psychotherap*”)] AND [(“Spain” OR “Spanish”) OR (“Japan” OR “Japanese”)]. We included articles published in peer-reviewed academic journals, written in English, Spanish, and Japanese between 2013 and 2023 that presented, designed, implemented, or assessed psychological interventions focused on suicidal behavior. (3) Results: 46 studies were included, concerning prevention, treatment, and training interventions. The risk of bias was low in both Spanish and Japanese studies, despite the lack of randomization of the samples. We identified common characteristics, such as psychoeducation and coping skills. Assertive case management was only highlighted in Japan, making an emphasis on active patient involvement in his/her care plan. (4) Conclusions: The findings will help professionals to incorporate into their interventions broader, more comprehensive approaches to consider more interpersonal components.

## 1. Introduction

Suicide continues to be the main cause of unnatural death in Spain. Despite how, historically, Spain has had the lowest suicide rates in Europe, in recent years this trend changed, increasingly alarmingly, reaching in the year 2022 its highest suicide rate of 4227 people. Among the highest risk groups, men between 30 and 39 years old stand out as well as youngsters between 15 and 29 years old. The most recent data to date in Spain, for the first half of 2023, show 1967 deaths (75.2% men and 24.8% women) [1].

Although suicide has been a topic of growing interest in recent years, in Spain there is still no national plan for its prevention, despite the need expressed by health professionals [2]. An example of a country that does have a strategy for its prevention at the national level is Japan [3]. Although it is not the only one that has it, there are also countries within the European Union such as Norway [4] or Finland [5]; Japan also has a long trajectory in the study of this phenomenon, also due to its high incidence [6]. However, Japan managed to reduce its suicide rate to historical values a year before COVID-19, with a total of 20,169 deaths in 2019. That decline began in 2012, with total suicide rates of 21.60, but despite this it increased again after the incidence of the pandemic, especially among youngsters (479 teens in 2020) [7]. The suicide-incidence data in Japan show a slight rebound (21,881 people in 2022), reaching values similar to those of 2016 with 21,897 people, although it is not as high as many years ago [8]. Last year, in 2023, it decreased by 63 from the previous year (21,818) [9]. It still impacts, as usual, the male sex, due to factors such as unemployment, which made this population more vulnerable, since a relationship was seen between unemployment and suicide rates [10], and despite COVID-19’s impact, it was more severe in the daily life of women [8].

From a psychological approach, interventions are defined here as those focused on suicide risk assessment, clinical treatments, and professional training in suicide-related practices [11]. A study that described the elements of interventions for suicide prevention plans across various regions in Spain showed that approximately 82% of them incorporated enhancements in early detection, assessment, prevention, and intervention for situations involving the risk of suicide. Regarding training interventions, there were found to be awareness-raising measures, guidance, and training workshops for effectively addressing the topic of suicide in the media and training specifically designed for professionals in sectors like education. It should be noted that the Valencian Community stood out as the sole region that specified a time frame within its objectives, considering that not all communities had a suicide-prevention program [12].

This highlights a need for consideration of the critical elements in suicide intervention and for identifying countries with extensive experience in addressing this issue, which could offer valuable guidance for our endeavors. In the inaugural nationwide analysis aimed at assessing the effectiveness of a national Japanese suicide-prevention strategy in various regions, including Japan and other countries, the study investigated the influence of the national fund in establishing frameworks and executing the nine proposed initiatives for suicide prevention. The findings determined that funding support at the national level for the implementation of suicide-prevention measures at the local level was an important factor, so that these could be put into operation, adapting them to the needs of the population and their capacity for implementation [3]. The introduction of a national suicide-prevention policy prompted a rise in the adoption of Gatekeeper Training (GKT) programs in Japan, indicating that the policy announcement resulted in favorable developments in the implementation of these programs that represent a fundamental element within suicide-prevention strategies in this country [13]

In addition to intervention programs in any of their modalities in Japan, the importance of educating about the issue of suicide has also been studied. Kodaka et al. [14] investigated the existing status of incorporating suicide education into undergraduate social-work programs in Japan. The study focused on examining the perspectives and apprehensions of educators. Over 80% of the survey participants expressed consensus on the importance of integrating suicide education into the academic curriculum for Clinical Social Work (CSW) and Psychiatric Social Work (PSW) students. Additionally, around 70% of those responsible for teaching CSW or PSW courses expressed a desire to incorporate topics related to suicide in their classes.

The government of Japan outlined objectives to reduce the suicide mortality rate, including enhanced support for practical initiatives at the local level (prefectures and municipalities), strengthening prevention measures for youth suicide, since it represents a particularly vulnerable population, and addressing suicide linked to work-related issues [15]. In different regions of Japan, innovative initiatives have underscored the significance of community-centric approaches to prevent suicide and intervene with high-risk individuals [16].

Therefore, given the high prevalence of suicide rates in Japan, its culture and attitudes toward this issue, its well-developed and innovative healthcare resources that create a supportive environment for the implementation and assessment of different suicide-prevention interventions and strategies, and its ongoing social and economic changes, which facilitate comparative research, we considered Japan a relevant and appropriate country for conducting a comparative study on suicide.

The aim of this study was to make a comparative review of what is published in scientific databases about the characteristics of psychological interventions directed to suicidal behavior or with an effect on it, designed for the Spanish and Japanese population with a scientific basis. Psychological intervention includes processes like health promotion, prevention, and treatment applying psychological principles and techniques by a health professional. Secondly, the aim was to find points in common and significant differences so that it serves to mutually apply in both countries to factors that may not have been considered and can benefit from their implementation.

Our research questions (RQs) were the following:-RQ1. What kind of elements were included in the interventions in each country?-RQ2. What prevention elements were included in Japan, but not in Spain?

Specifically, we aim to do the following:-Identify and analyze the specific elements included in suicide interventions implemented in Japan and Spain;-Compare and contrast the suicide-prevention elements that were part of the interventions in Japan and that were not present in the interventions in Spain;-Identify successful or innovative practices that could be adapted or implemented in the Spanish context in order to be an example to follow for improving suicide-prevention strategies.

Focusing on the period of 2013–2023 and covering almost the beginning of the decline of Japan’s high suicide rates, we offer a critical vision of the psychological approach to this problem, bringing to light aspects that may have been ignored in both countries and that may have been key to the intervention, thus benefiting them reciprocally from these findings, although always with a certain margin of variability, considering cultural differences.

## 2. Materials and Methods

### 2.1. Information Databases and Searches

The PRISMA verification protocol was used [17] for the development of reviews for both countries. A literature search was conducted in the electronic databases PsycInfo, Web of Science, and Scopus due to their specialized coverage in the field (PsycInfo), multidisciplinary scope (Web of Science, Scopus), and advanced search and citation analysis features, facilitating systematic reviews. We used the following terms: [(“suicide” OR “suicidal behavior” OR “suicidal attempt” OR “suicidal thought” OR “suicidal intention”) AND (“prevention” OR “intervention” OR “psychosocial treatment” OR “Dialectical Behavior Therapy” OR “Cognitive Therapy” OR “psychotherap*”)] AND [(“Spain” OR “Spanish”) OR (“Japan” OR “Japanese”)]. The search covered articles from 2013 to 2023 written in Spanish, English, or Japanese, and was made on 9 June 2023 on PsycInfo, Web of Science, and Scopus, and then on 20 August 2023, adding the databases PsycArticles and PubMed. Finally, the search in PsycArticles was eliminated since the results were much fewer than in the other databases and did not provide documents different from those already found in these.

In relation to the search strategy, we carefully selected keywords related to suicide, including various aspects such as suicidal thoughts, so that it gave a wide range of results, as well as terms associated with prevention and intervention strategies. The use of the Boolean operators “AND” and “OR” allowed us to obtain a structured and focused search query. By using “AND”, we ensured a refined search of the literature specifically addressing suicide interventions. Meanwhile, “OR” broadened the search to include synonyms and related concepts. 

Our choice of search terms may not encompass every possible aspect of suicide-prevention interventions, and there was a possibility of omitting relevant studies due to variations in terminology or indexing practices, although we endeavored to include a diverse range of keywords to mitigate this limitation.

### 2.2. Data Collection

A search using Mendeley Desktop was made two times, in May and in September of 2023, for this comparative review, and then the screening of the documents. Mendeley Desktop allowed for reference management, collaboration between authors, and integration with word processors. The lead author of this review, N.L.M.-R., made the initial search independently, so the references of the documents were extracted from the databases. The title and abstract screening of all papers was independently performed by two of the authors (M.d.P.M.C. and N.L.M.-R.) with reference to the inclusion/exclusion criteria. Those articles that in their title included the word review/meta-analysis, non-suicidal self-injury, names of populations or countries other than Spain and Japan, and interventions focused on treating suicide grief were discarded. Then, reading the abstract, we could remove articles that tried to validate tests, models, or psychological theories, as well as those that included non-psychological interventions. There were two disputed studies that seemed to be the follow-ups of other studies included in the final sample. This was resolved through discussion between two authors (M.d.P.M.C. and N.L.M.-R.). After full-text reading, we were able to compare the metadata of the studies to clarify the presence of articles that could be the progression of others, such as the article with the follow-up results of another study in the group of Spanish articles, and another independent in the Japanese intervention group. Finally, using tables, the characteristics of interest of the articles (main objectives, samples, aspects addressed, places of action, duration, components of the interventions, and their phases) were revised for all the authors.

### 2.3. Assessment of Risk of Bias and Reporting Quality

In addition to conducting a thorough search, four members of the group (N.L.M.-R., M.d.P.M.C., Y.K., and B.D.) independently reviewed the full-text screening process and its outcomes. In instances of disagreement, we collaboratively revisited the items in question, aiming to identify specific features that could warrant their exclusion or inclusion. Through this joint effort, we reached a consensus that determined the final selection of items. As for the articles written in Japanese, the member of the group originally from this country, Y.K., carried out a more exhaustive review of these.

To assess the risk of bias in the gathered studies, we dissected and presented the interventions’ characteristics, considering some relevant criteria on the detailed reporting of manuscripts: (a) criteria for patient inclusion; (b) the control or comparison group; (c) treatment descriptions; (d) characteristics of the sample; (e) data on outcomes; and (f) the inclusion of lost-to-follow-up patients. The first author, N.L.M.-R., coded each item as either met or not met (including cases where it was not clear) for each document. After this, the second author, M.d.P.M.C., supervised this process.

The assessment process in evaluating the risk of bias was structured around key components, through a tailored approach aligning with established criteria outlined in tools such as Cochrane Collaboration’s Risk of Bias Tool [18] for randomized controlled trials or the Newcastle–Ottawa scale [19] for non-randomized studies. 

Since the focus of analyzing the studies was to compare the common characteristics of interventions conducted in both countries and their outcomes, , a narrative synthesis of the findings was used. In this way, we tried to identify and analyze specific elements included in suicide interventions in Japan and Spain, such as types of interventions and prevention elements; to compare elements in Japanese interventions absent in Spanish interventions; and to identify successful or innovative practices from Japan for adaptation in Spain, considering implementation methods, demographic characteristics, and contextual factors.

### 2.4. Study Selection

Articles were included if they (a) presented, designed, implemented, or assessed psychological intervention programs focused on suicidal behavior, or whether the intervention, although not primarily directed at suicidal behavior, influenced this; (b) the interventions were designed for or applied to the Spanish or Japanese population; (c) or they were published in a peer-reviewed academic journal, written in English. Articles were excluded if (a) the study was a review or meta-analysis, (b) the study tried to validate tests, models, or psychological theories, (c) they were focused on populations other than the Spanish or Japanese, or they were addressed to ethnic minorities living in other countries, (d) they were other formats different from the scientific article, or (e) they included interventions in modalities other than psychological.

There were no restrictions on the design of the studies since this was also an evaluable aspect.

The period between 2013 and 2023 was chosen since 2012 was the turning point from which a decline in suicide rates was seen in Japan, although recent years were also considered despite a new rise in ratios as a result of the COVID-19 pandemic to give a broader and more current overview of the problem.

### 2.5. Data Analysis

Given the variability in interventions focused on the problem of suicide or risk variables related to it, what was sought were not numerical values, but rather the analysis of the qualitative characteristics of these interventions, as well as the results of their effectiveness. To address this objective, we conducted a content analysis to consolidate and synthesize the results following the Braun and Clarke’s six steps [20]. Microsoft tools such as Word and Excel were used for manually organizing and analyzing textual data extracted from included studies. The process began with an initial review of key concepts in titles and abstracts to filter relevant articles (terms derived from suicide or psychological interventions were applied for this problem). After familiarizing ourselves with the data, we found the following themes: training programs, prevention, and treatment interventions. After that, we identified initial codes (intervention components) for articles related to our interest in interventions affecting suicidal behavior. Extracting relevant information from selected articles involved a specific search for key themes and subtopics, followed by a deeper review to determine refinements. Ultimately, we adjusted and clarified potential themes and subthemes to address our research questions investigating the initial framework from the introduction of the studies and the results observed both in the short and long term. The subtopics included aspects on which it intervenes and the framework or approach with which it is applied.

## 3. Results

The initial bibliographic search yielded a total of 1430 outcomes including the terms “Spain” or “Spanish”, compared to 1258 outcomes including the terms “Japan” or “Japanese”. Following the removal of duplicates, 883 articles were chosen in the Spanish search and 629 in the Japanese. Abstracts were then examined to identify studies for inclusion, and findings were extracted from the complete texts. After applying inclusion and exclusion criteria and scrutinizing the title and abstract of each study, 23 studies were ultimately included in each review, making a total of 46 articles examined. Figure 1 and Figure 2 display the PRISMA Flow Diagram, illustrating the process of article selection made for both countries.

### 3.1. Methodological Quality Analysis

The analysis based on the criteria already presented showed that not all the interventions had a control group to compare their effects. A proportion of 4/23 in the Spanish sample [21,22,23,24] and 5/23 in the Japanese sample [25,26,27,28,29] included a control group exempt from treatment, although some had a group that received a different treatment, with a proportion of 5/23 in the Spanish sample [30,31,32,33,34] and 3/23 in the Japanese sample [35,36,37]. While, in the Japanese sample, two articles [38,39] included the relatives of the patient, in the Spanish sample we found two articles [24,40].

All the interventions, both in Spanish and Japanese samples, included a description of the treatments and data on outcomes. Regarding the inclusion of lost-to-follow-up patients, all the Spanish articles included this, except that of Espandian et al. [41], since it is focused on effective strategies in a specific pandemic context, Marco et al. [40], because it is a randomized control trial, and Reijas et al. [33], due to its retrospective nature. And, in the Japanese sample, we found more articles without lost-to-follow-up participants [26,28,35,42,43]. In both samples, most studies were non-randomized for the convenience of accessible patient groups, except for four studies in the Spanish sample [24,34,44,45] and eight in the Japanese sample [37,38,46,47,48,49,50,51]. Finally, as to the characteristics of the samples of the articles, they are explained in detail in the following section.

### 3.2. Sample Characteristics

The samples size of the Spanish articles varied widely, from 30 to 12,596 participants, and was the same in the Japanese sample, ranging from 19 to 631,133 participants.

Regarding gender, there is a prevalent trend where, despite the inclusion of both genders in the samples, the proportion of women is noticeably greater than that of men. This aligns with the existing literature indicating that women exhibit a higher attempt rate than men, despite men being more prominently represented in completed suicides.

### 3.3. Study Characteristics

Eligible studies were published between 2013 and 2022 with a median = 2019 on the Spanish sample and median = 2017 on the Japanese sample.

The interventions described in the studies included the following: (1) face to face and telephone contacts, (2) telephone consultations, (3) telephone follow-ups, (4) smartphone-basedI interventions, (5) group interventions, and (6) other technology-based interventions. Very brief interventions were found in both samples, from a 2 h training session to months exceeding a year in duration.

Regarding the interventions, there are various interventions including those cognitive, cognitive–behavioral, supportive counselling, dialectical behavior therapy, educational interventions, and humanistic therapies. We observed a greater number of case interventions, although the sample number of the articles was high, more than community or group interventions, in both groups of articles (Spanish 4/23 and Japanese 7/23). Although, there may be some that combine individual treatment with group treatment [24,34,35].

With regard to the components of the interventions, in both samples the following can be seen: psychoeducation about anxiety, depression, and suicide; practical training (role play); coping/problem-solving strategies; emotional regulation strategies; mindfulness; cognitive restructuration; behavior activation; relaxation techniques; visual information through videos or written messages; anxiety-control strategies; mentalization exercises; and the validation and humanization of the collective experience. But the Japanese sample differs in the inclusion of discussion groups, assertive case management, and gatekeeper training.

With respect to the time duration of the interventions, in the Spanish sample, we found shorter interventions focused on treatment once the problem had already appeared, lasting a few hours/weeks. However, longer interventions ranging from months to years were more focused on preventing damage. In the Japanese sample, we found longer interventions in treatment, , and brief training interventions of a few hours, even though these are more numerous in this sample. In interventions aimed at prevention, we found more variability in proportion, from brief interventions of a couple of hours (3/11) to longer ones over months (4/11) or years (4/11).

#### 3.3.1. Training Programs for Other Professional or Non-Professional People in Its Detection and/or Treatment

In the Japanese sample, we observed many articles of this type, nine, compared to the Spanish sample with only one article about a training intervention. The age of the sample of the Spanish training program is not specified, but on the Japanese interventions, three are focused on young adults and the rest on middle adults. Regarding the sites where they were carried out, three were on educational institutions, three on clinical settings, two were carried out online, and one in a local government.

#### 3.3.2. Health Promotion and Suicide-Prevention Interventions

In the Spanish sample, we found a total of 11 suicide-prevention interventions, similar to the Japanese sample with 10. Of the 11 Spanish interventions, 4 were focused on adolescents, and the rest were adults with a mean age around 40 years. Compared to the Japanese sample, where we found more variety in targeted age ranges, three interventions were focused on minors, three on young adults (between 22 and 32 years old), one on middle-aged adults, one on elderly people, and three included people of very different age ranges, from teenagers to older adults.

Most of the Spanish interventions were designed and/or developed for clinical/medical settings, except one of them that was outpatient [52] and another that took a double route, online–telephone [53]. On the other hand, in the Japanese sample, four of the interventions took place in clinical settings, three in educational institutions, one online, one promoted by the local government, and another in a residential care setting.

#### 3.3.3. Treatment of Suicidal Behavior Interventions

Eleven of the total Spanish sample of articles were about the treatment of this problem, although there were actually 10 interventions, since two of the articles evaluated the effectiveness of the same treatment in the short and long term, a significant amount compared to the Japanese sample, with 4 interventions. Most of the Spanish interventions were focused on adults, and the samples that included minors also included older age ranges, so they did not focus on youngsters specifically. The Japanese interventions were focused on young people older than or equal to 20 years.

In the Spanish sample, like prevention interventions, most of these were designed and/or developed for clinical/medical settings, except two that were outpatient [22,54] and one in a residential setting [55]. Regarding the Japanese interventions, a total of four interventions were developed in clinical settings.

Table 1 and Table 2 describe the main features of the selected articles with the terms Spain or Spanish, and Japan or Japanese, correspondingly.

A more detailed analysis of the effectiveness-proven Spanish and Japanese interventions is shown in Table 3.

To summarize, among the main communalities in the Spanish and Japanese interventions focused on prevention or treatment, we highlight a component of psychoeducation, even it is brief [40,51], coping skills [26,30], the promotion of adherence to treatment [38,62], and key concepts about anxiety and/or depression [37,53], but we can also find this component of psychoeducation about depression in training interventions, like Hashimoto et al. [63] in the Japanese sample. We find some peculiarities in Spanish interventions strategies, like validating and humanizing the collective experience [53]. As for the differences, these are more evident in the training programs; the Japanese interventions offer more specific and complete information about suicide, such as warning signs of suicide and risk factors [25], and the Spanish one includes more general strategies, like interpersonal skills or communication-skills training [40].

## 4. Discussion

This systematic review aimed to identify suicide-prevention interventions in Japan and Spain and compare them. From 2688 articles (1258 in Japan and 1430 in Spain) retrieved, 46 papers (23 in Japan and 23 in Spain) were included. The studies examined in this current review delve into the provision of information and counselling for academic and clinical settings to improve or implement measures regarding the problem of suicide. The objective is to compare the most recent published interventions in both countries, recognizing underlying problems and factors influencing that behavior, as well as possible effective strategies against suicide, encouraging their involvement in this problem. There are notable differences among the studies in terms of intervention types and the evaluation of measured variables. This divergence complicates the comparison of outcomes. Although, an important strength in this review was the variety of the methodological frameworks of the included studies, such as quasi-experimental studies or observational studies, since this offers valuable insights, being cautious in their interpretation and able to mitigate potential confounding factors to ensure robust findings.

About the feasibility of the interventions, most of them were feasible to implement, and focusing on the records of follow-ups, they were able to retain participants possibly due to the use of stable contexts, such as clinical settings or educational institutions.

As for key similarities and differences between Japanese and Spanish interventions studied, we can verify the following:

Similarities:-The prior or subsequent training of those in charge of the interventions;-Greater complexity/completeness in remote interventions to compensate for the lack of contact;-A proactive approach in the deployment of interventions.

Differences:-Spanish interventions focused on short-term outcomes;-Most of the interventions dedicated to prevention in Spain are still under evaluation, whereas Japanese interventions showed more contrasting results with long-term positive outcomes;-Japanese Interventions covered larger groups and mixed age ranges.

### 4.1. Intervention Types

Consistent with the literature about suicide, which advocates for prevention rather than postvention [12,70,71], we mostly found, in both countries, that interventions aimed at prevention and the training of professionals to carry it out on vulnerable populations (e.g., young people and hospital patients), allowing them not only to preserve their health in environments that are also habitual for them, but also to involve those responsible on many occasions for the well-being of these population groups in these controlled environments. Stable contexts allowed for collaborative partnerships, a greater control of participants, and facilitated long-term follow-ups, thus increasing the chances of success for interventions.

Answering the research questions, regarding the kind of components that were included in the interventions in Spain and Japan, we can mention those that included psychoeducation about the problem, gatekeepers, techniques to improve communication (assertiveness, help-seeking and role play), and coping strategies and techniques for managing anxiety and depression, like relaxation exercises or behavioral activation. Specifically, among the innovative effective elements included in the Japanese interventions that have not been found in those implemented in the Spanish interventions, we should highlight such a direct approach when the medium used is remote (e.g., online with encouragement messages), when leadership has involvement with messages to the community and the promotion of social support networks between them, and the use of assertive case management. The involvement of leaders is essential, as we can see that public campaigns on this topic already found good results before in Japan, implemented in specific areas [72]. To some extent, these results are consistent with studies that support these types of strategies in suicide intervention [73,74,75].

As we can see in this review, prevention strategies utilizing emerging technologies were used in interventions in recent years, especially in the Japanese sample, although research in this area remains limited [76]. The effective utilization of technology in suicide prevention still presents a significant challenge [77], teaching both patients and healthcare professionals to use it.

At an economic level, we know the impacts these changes can have on suicidal behavior and even more so in Japan, being more sensitive than social factors [78]. It should be noted that the three interventions that considered the COVID-19 situation and/or were developed after this event [41,53,55] within the Spanish sample were focused on prevention, which already denoted a growing interest in treating this problem, possibly aggravated by the crisis caused by the pandemic. In the Japanese sample, only one intervention included COVID and focused on treatment [25]. From the start of the COVID-19 pandemic in 2020, we did not find more published interventions for suicide compared to the rest of the years. In view of our results, we cannot consider 2020 a year from which COVID will take on great relevance in the publication of interventions for this problem, perhaps considering these last years as a stable phase in the implementation of interventions for this problem, since we found other articles after this year that do not mention it. This fact fits with the data about a decline in suicide rates during the early months of the pandemic in Japan [79,80], although the negative effects may not have been evident so early, since later, the COVID-19 pandemic showed a negative impact on suicide rates in Japan, especially in women and youngsters [81].

Despite the challenges posed by external factors such as the COVID-19 pandemic, individuals and communities developed resilience and coping mechanisms that influenced the effectiveness of suicide interventions. Given that many of the interventions became remote, new ways were sought to create connection despite the lack of non-verbal information, and the provision of interactive material through technology seemed to help adherence, in addition to the flexibility and security that these means offered.

### 4.2. Cultural Considerations

Although eastern-Japanese culture prioritizes stable interdependence, this is in contrast with western cultures, like Spain, that emphasize individual independence and maintaining familial ties [82]. Social support was a variable that was highly taken into account in the interventions aimed at both the Japanese and Spanish populations, either including their relatives in the intervention or facilitating contact with them while they were being treated, as poor social support is a very important variable that predicts suicide attempts, especially between youngsters [83].

Socio-cultural differences between Spain and Japan, including attitudes towards suicide, interpersonal needs, disclosure practices, and personal values, should be explored to understand their impact on suicide rates and intervention effectiveness. By contextualizing these factors, researchers can develop interventions tailored to each population’s unique needs. Examining their norms, beliefs, cultural values related to individualism vs. collectivism, and attitudes towards suicide, we will be able to predict the acceptability and efficacy of intervention strategies in each country.

### 4.3. Limitations

On the one hand, ensuring that published articles meet certain standards of quality and reliability, we were limited to scientific databases for research purposes, and we did not consider public documents from central and local governments and the grey literature [84]. By focusing on articles published in these journals, our systematic review was more likely to include studies that had undergone thorough scrutiny by experts in the field, in addition to its greater power of impact and dissemination outside the origin country.

On the other hand, there were psychological-intervention programs that were combined with other treatments in another modality, for example, pharmacological, which made it difficult to discern what degree of effectiveness was attributed to the psychological program. We also did not evaluate what parts of the interventions were effective [76]. Finally, a limitation that comes from the sample used in some interventions refers to its size or its origin, since, for example, with very small or non-randomized samples, it is difficult to make a generalization.

## 5. Conclusions

Overall, similarities between Spanish and Japanese interventions focused on prevention or treatment included elements such as psychoeducation, coping skills, or the promotion of treatment adherence. However, Spanish interventions emphasized interpersonal aspects, while Japanese interventions are more specific and comprehensive to the issue.

Regarding the characteristics of the interventions, we could highlight the shorter duration of those carried out in Spain, focusing on immediate treatment, while Japanese interventions span longer terms, cover bigger groups, and comprises more varied age ranges.

These differences underscore the need for tailored suicide-prevention policies in both countries. Spain may benefit from incorporating broader, comprehensive approaches akin to those in Japan, while Japan could consider integrating more interpersonal components into its interventions. Additionally, the longer duration and broader age inclusivity of Japanese interventions suggest a more comprehensive strategy for suicide prevention, offering valuable insights for shaping policy decisions to reduce suicide rates in both Spain and Japan.

It is important to carry out this type of comparative work so that the position of the countries of interest in terms of mental health problems can be seen Suicide is a problem that affects developed and developing countries to a greater extent. Therefore, countries like Spain that do not have a National Strategic Plan for Suicide Prevention can take the example of countries like Japan that do, and have been very well received, despite the stigma of this problem. It also shows that, despite the cultural differences that may exist between countries such as Spain and Japan, there are common aspects that can be useful in both countries.

Although Japan would not be the only country that has had a long history with this problem and significantly reduced its suicide rate, it does share with Spain a growing concern about the rise among youngsters. The OECD presented Japan as one of the countries with the highest suicide rates among under-30s [85].

In addition to the importance that cross-cultural studies have, applied to test psychological models and theories in different cultures, they are also very useful for studying cultural and psychological variations that may or may not be present in our own cultural experiences. In the context of cultural adaptation, if we wanted to incorporate in Spanish intervention elements that have been effective in the other country and vice versa, research should explore the integration of cultural elements and understanding into treatment approaches, since significant differences can be found in models of training and service structures, and also, of course, elements that can be effective in one society, given the values, lifestyles, and education, which do not have to coincide with those of another.

Despite the existence of cross-cultural studies that explore the different perspectives in addressing different problems [86], there are no studies that focus specifically on comparing the different interventions developed in both countries in such an exhaustive way.

We believe in the importance of this kind of research, especially influencing the policies in the prevention and treatment of suicide in both countries, as it serves as an example of mutual learning.

In summary, comparative studies, even when they are between two specific countries, like Japan and Spain, can provide valuable information that other countries can use to develop more effective suicide-prevention strategies, identifying risk and protective factors that possibly go ignored, improving mental health services’ analyses of disparities in their accessibility and quality, and addressing the specific needs of different cultural groups within their populations (attitudes toward suicide, mental health, and help seeking).

### Future Directions

Cultural factors like the perceptions of suicidal Spanish population groups have could be found between the implications for policy development, as well as the feasibility and effectiveness of community-based suicide-prevention initiatives.

Longitudinal studies will continue to give us clues about what the future objectives are and how the population needs are changing, while ensuring certainty about those aspects that remain effective over time.

It is necessary to continue learning and taking examples from other countries that have already found themselves in similar critical situations, while taking into account cultural differences and adapting them to the target population. It is expected that these types of comparisons encourage governments to implement a national suicide-prevention plan that is in high demand, since prevention is much more important than postvention.

Globally, suicide has become an alarming threat to society. All healthcare practitioners have a responsibility to disseminate awareness and information among the population about the measures that can be taken for its prevention. Strategies more focused on suicidal-behavior information were most frequently addressed in the studies carried out in Japan and more interpersonal approaches in Spain. This attempt could help societies to be aware of the clinical characteristics that are most relevant in this approach.

## Figures and Tables

**Figure 1 healthcare-12-00792-f001:**
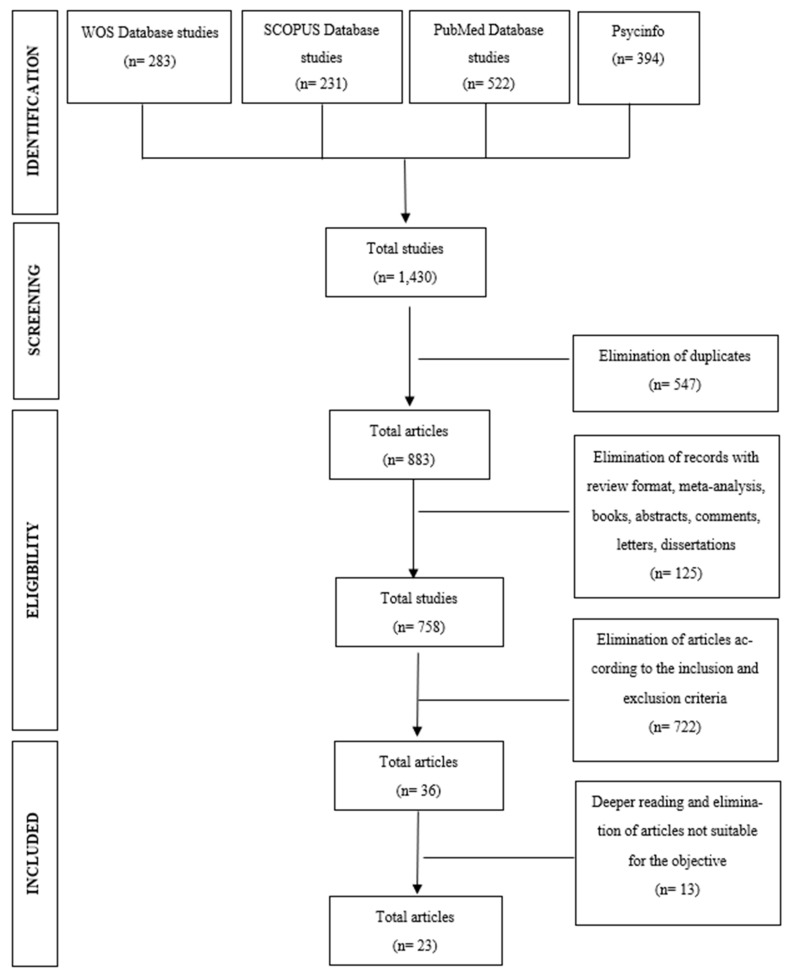
Flowchart of the selection process of the articles of review with the terms Spain or Spanish, following PRISMA.

**Figure 2 healthcare-12-00792-f002:**
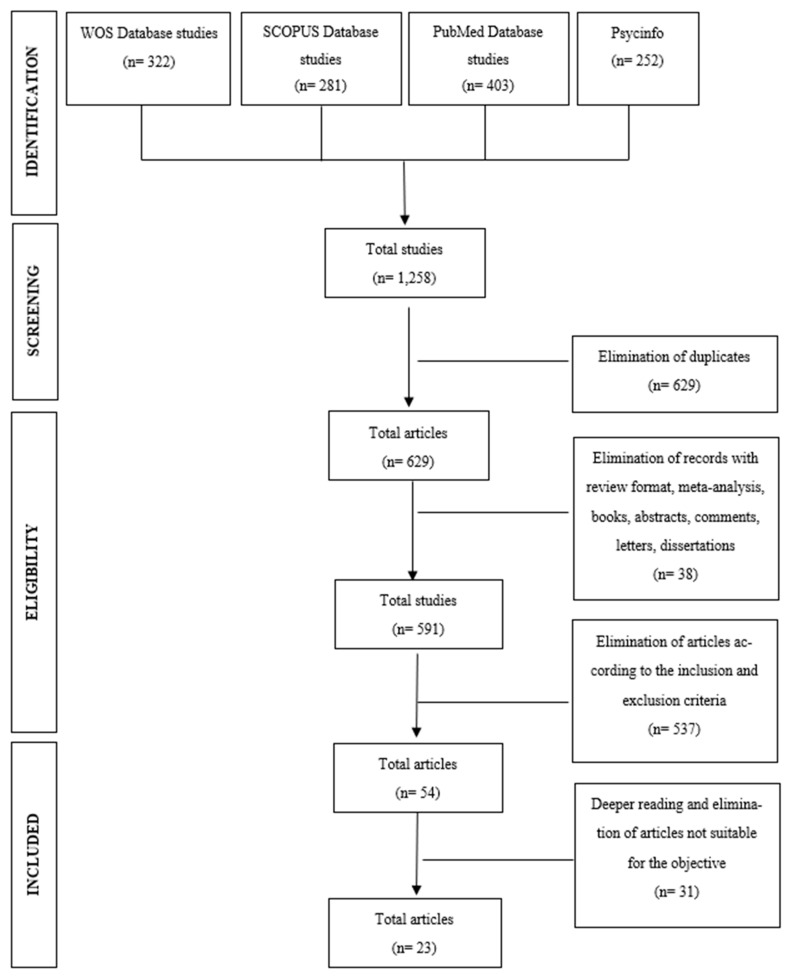
Flowchart of the selection process of the articles of review with the terms Japan or Japanese, following PRISMA.

**Table 1 healthcare-12-00792-t001:** Main characteristics of the interventions in the Spanish population.

Study	Main Objective	Target Population	Aspects on Which It Intervenes	Place of Action	Duration	Components	Phase (Proven Effectiveness)	Modality	COVID Inclusion
Albuixech-García et al. (2020) [21]	Treatment. Examine sociodemographic factors linked to suicidal behavior and assess the impact of a nursing care protocol on follow-up for such patients.	213 participants (13–91 years old); intervention group (51.6%); and control group (48.4%).	Suicidal behavior and related factors.	At hospital.	2 h training session about the protocol.	- Usual discharge protocol: written emergency-department discharge letter. - Mental health care continuity-chain protocol. Before executing the protocol, every nursing professional received a training session that comprised a theoretical component which explained the action plan design, and a practical section.	Quasi-experimental study.	Face to face.	No
Angora et al. (2022) [30]	Treatment. To evaluate an intensive suicide-reattempt-prevention program using brief problem-solving therapy in conjunction with a case-management approach.	871 patients (292 received treatment as part of ISRPP and 357 received treatment as usual (TAU)).	Suicide reattempt.	In clinical settings.	Eight weekly sessions.	Psychological assessment, explanation and planification of the intervention (discussion of different ways of coping with suicidal behavior, strategies for the management of suicidal thoughts, problem-solving, and focus on aspects of relapse prevention).	High cost-effectiveness.	Face-to-face consultations and telephone follow-ups.	No
Barrigon et al. (2022) [54]	Prevention. To evaluate the effectiveness of a smartphone-based Ecological Momentary Intervention to prevent suicidal thoughts and behavior.	Patients older than 18 years old.	Suicidal thoughts and behavior.	Outpatient intervention.	12 months.	Safety plan with coping strategies, family contact options, relaxation videos, pre-recorded messages, health resource links, and emergency services. Enhanced app intervention with personalized well-being messages and information on accessing preferential or urgent care. Mental toolbox with relaxation, behavioral activation, and mentalization videos.	Still unproven effectiveness.	Smartphone-based.	No
Bergmans et al. (2021) [53]	Prevention. To show the lessons learned by The Skills for Safer Living (SfSL) team in shifting a comprehensive 20-week in-person intervention to a virtual model.	A group that includes all genders over 18 years old who have attempted suicide one or more times.	Key concerns about suicide and strategies and skills to improve mental health.	Online (Microsoft Teams) and telephone support.	20 weeks.	Key concepts, strategies, and skills for common concerns (anxiety and suicidal ideation). Strategies: validating and humanizing the collective experience.	Pilot phase.	Group in-person intervention to a virtual model.	Yes
Cebrià et al. (2013) [56]	Treatment. To determine the effectiveness over one year of a specific telephone management program on patients discharged from an emergency department after a suicide attempt.	296 patients without age limit.	Time elapsed between initial suicide attempt and subsequent one, changes in the annual rate of patients who reattempted suicide.	Emergency department of a hospital.	12 months.	Brief interview to re-assess the risk of suicide. Ordinary 5–10 min telephone follow-up, detecting significant changes. And 15–45 min interventions for situations of crisis. In some cases, follow-up was arranged with a primary physician.	Results confirm the effectiveness.	Individual telephone format (fixed line and mobile telephone).	No
Cebrià et al. (2015) [57]	To determine the effectiveness over 5 years.	296 patients without age limit.	Time elapsed between initial suicide attempt and subsequent one.	The emergency department of a hospital.	12 months.	The experimental intervention included a series of measures to increase adherence to usual treatment and brief interventions in situations of crisis.	Limited long-term effectiveness.	Individual telephone format.	No
Espandian et al. (2021) [41]	Prevention. To carry out interviews and interventions for patients with suicide risk and substance-use disorders.	Patients with suicide risk and substance-use disorders.	Suicide risk.	In clinical settings.	Not specified. But recommend established time periods.	- Decisional balance; - Crisis-stabilization/intervention plan; - Activities that involve experiencing positive emotions; - Emotional regulation strategies and abilities to handle problematic thoughts or ideas; - Improvement of the adherence to health services.	Limited research proving their effectiveness.	Face to face and by telephone.	Yes
Fernández-Artamendi et al. (2019) [58]	Treatment. To evaluate the differential effectiveness of the combination of various strategies for the prevention of the repetition of suicide attempts.	163 patients with an age range between 18 and 80 who attended emergency services after a suicide attempt.	Suicide reattempts.	Emergency department of a hospital.	30 months.	Interventions: - Passive treatment (information leaflet about prevention of suicidal behavior); - Active case-management treatment module with regular interviews, promotion of treatment adherence, and contact with available social resources; - Program of psychoeducation: communication skills, coping skills, analysis of psychological changes leading to high-risk situations, risk and protective factors, stress management and understanding of the role of social support and health services.	No significant differences between the three groups in the number of suicide attempts nor in the number of patients with more than one suicide attempt.	Face to face, but on the phone if necessary.	No
Gabilondo et al. (2020) [22]	Treatment. To analyze the results of a 6-month telephone follow-up program for the prevention of suicidality in adult patients after a suicide attempt.	Adult patients (average age: control group 45.2, intervention group 41.2) treated in hospital emergency departments following a suicide attempt and subsequently discharged after a psychiatric evaluation.	Evaluate the current risk of suicidal behavior; reinforce adherence to treatment and follow-up from a healthcare professional; contribute to psychoeducation; and carry out a crisis intervention in case of emergency.	Home.	6 months.	Psychoeducation; follow-up calls; and crisis intervention in case of an immediate risk situation.	Short-term intervention proved as effective, but not long-term.	Telephone.	No
Gomes-da-Costa et al. (2021) [59]	Prevention. To describe the CSRC preventive strategy experience in a tertiary hospital in Barcelona.	365 patients (59.7% female), mean age 44.9 ranging from 18 to 92 years.	Suicide risk.	At a hospital clinic.	12 months.	Three phases: 1. Alert and activation phase; 2. The psychiatrist completes the MINI suicidal module from a MINI interview; 3. Follow-up phone call and an appointment in the outpatient clinic.	CSRC protocol reduced hospitalizations and the mental healthcare utilization in the first year after discharge from the psychiatric emergency room.	Face to face and by telephone intervention.	No
Jiménez-Sola et al. (2019) [52]	Prevention. To evaluate the suicide-risk prevention program ARSUIC by estimating the degree of implementation, fulfilment, and effectiveness.	1633 patients (mean age 39.08–42.85), who received medical and mental healthcare at the emergency department due to a suicide attempt.	Time between discharge and the first outpatient visit, proportion of suicide reattempts, attempt rate per person—year and time between attempts.	At the hospital.	Without specifying, an appointment with an outpatient psychiatrist.	Appointments with a psychiatrist without training in suicide prevention within a maximum of 7 days following discharge, plus their outpatients’ usual appointments.	Reduced the time between discharge after a suicide attempt and the first outpatient appointment. Decreased suicide attempts due to reattempts and the rate of attempts per patient and year.	Observational, retrospective.	No
López-Goñi et al. (2021) [31]	Treatment. To analyze the sociodemographic and clinical characteristics of the TFP group and the routine treatment group; to compare the differences between both groups, as well as the recurrence of suicidal behavior.	Patients older than 18 years admitted for a psychiatric emergency. The first sample n = 207. In the second, n = 203.	Suicidal behavior.	Outpatient intervention.	Over 12 months.	Psychiatric specialists conducted assessments, explained the study protocol, and conducted interviews. A year later, a team member reviewed electronic records, collecting specified follow-up variables.	Contradicts the recommendation of the previous research to be applied on patients who had repeated SAs several times and not in first-time patients.	Telephone-based.	No
Marco et al. (2022) [40]	Training. Verify the efficacy of the Family Connections intervention.	124 participants, relatives of people diagnosed with suicidal behavior disorders.	Psychological variables: burden, anxiety, depression, and quality of life.	Clinical settings.	12 2 h sessions once a week.	Psychoeducation, emotion-regulation training, interpersonal-skills training, communication skills training, and problem-solving training.	Results confirm the effectiveness.	Group, face-to-face format.	No
Martínez-Alés et al. (2021) [60]	Treatment. To determine the cost-effectiveness of two strategies for post-discharge suicide prevention, an enhanced contact intervention based on repeated in-person and telephone contacts, and an individual 2-month-long problem-solving psychotherapy program.	1492 patients aged older than 18 years old.	Post-discharge suicide relapse.	In each general hospital’s emergency department.	Three outpatient appointments and telephone follow-ups; 2-month program of weekly individual psychotherapy sessions.	Individual psychotherapy based on a problem-solving-therapy approach, and 15-min-long follow-up telephone calls at months 1, 6, and 12.	Cost-effectiveness analysis.	Face to face and telephone.	No
Martínez-Alés et al. (2019) [44]	Treatment. To evaluate the clinical effectiveness of an intervention at reducing the risk of relapse among patients discharged from the emergency department after a suicide attempt.	1775 patients, mean age 40.5, treated at a general hospital ED due to a suicide attempt.	Risk of relapse after a suicide attempt.	At an emergency department in a hospital.	A single appointment.	Follow up on single appointment within 7 days after discharge following a suicide attempt.	Confirm effectiveness.	Face to face.	No
Muela et al. (2021) [55]	Prevention. To describe a pilot study evaluating the Over Come-AAI program for preventing suicidal behavior.	30 adolescents aged between 14 and 17 years.	Suicidal behavior and non-suicidal self-harm, and improvements in indicators closely related to suicidal behavior (mental pain, hopelessness, and depressive symptoms).	In a residential childcare setting.	Six sessions.	Pretest (evaluation questionnaires), intervention: 1. Facts, beliefs, and myths about suicide, progressive muscle relaxation; 2. Risks and protective factors of suicide and respiratory energization; 3. Warning signs of suicide and diaphragmatic breathing; 4. Connectivity, self-pity, and negative criticism and mindfulness; 5. Suicide-risk safety plan and mindfulness; 6. Closure and additional help resources and focus on emotions post-test.	Pilot study.	Group-based.	Yes
Pérez et al. (2021) [32]	Prevention. To determine the incidence of suicide attempts in Spain and their main risk factors; and to explore the effectiveness of different secondary prevention programs, compared to treatment as usual.	2000 people (≥12 years of age) who have made a suicide attempt.	Suicidal behavior.	At the psychiatric emergency ward of public, general, or university hospital.	12 months.	SURVIVE trial involves two interventions: - Telephone-based management: Initial call to assess current suicide risk, follow-up for treatment information, adherence, and stressors. Crisis intervention for detected risk, with emergency clinic appointments if necessary. - iFightDepression Survive: Online cognitive–behavioral tool focusing on behavioral activation, cognitive restructuring, sleep regulation, mood monitoring, and healthy habits. For participants aged between 12 and 18 years old, self awareness of mental health addresses suicide risk factors, knowledge on depression and anxiety, coping skills for life events, stress, and suicidal behaviors.	Data will be obtained on the effectiveness of secondary prevention programs.	Face to face and telephone-based.	No
Pérez et al. (2020) [61]	Prevention. To describe the Catalonia Suicide Risk Code and its implementation.	12,596 patients (8077 females with a mean age of 40 years and 4519 males with a mean age of 43) with an activated CSRC.	Detect suicide risk and risk factors like the diagnosis of a life-time psychiatric disorder, stressful life events, and hopelessness.	The emergency department of a hospital.	Not specified.	Combined risk assessment, appointments with a mental health professional, telephone calls, follow-up care of patients at risk of suicide, and a neuropsychiatric interview (MINI) MINI suicidality module.	Early implementation.	Face to face and telephone.	No
Pons-Baños et al. (2020) [23]	Prevention. To identify the sociodemographic and clinical characteristics of individuals with suicidal behavior, and to analyze differences between non-participants and participants in a nurse-led prevention program.	753 adult patients (464 women) with suicidal behavior, mean age 43.44.	Suicidal behavior.	At hospital.	Over 12 months.	Coping strategies, crisis intervention, anxiety reduction, counselling, cognitive restructuring, and preventing substance abuse.	Effective intervention.	Face to face and by telephone intervention.	No
Reijas et al. (2013) [33]	Treatment. To evaluate the effectiveness in reducing repeated suicide attempts in the Intensive Intervention Program (IIP).	191 patients, 89 in the treatment group and 102 patients in conventional treatment group, mean age 39.63.	Suicidal relapse.	At hospital.	10 sessions (6 months).	Three phases, cognitive–behavioral therapy: - First phase: cognitive conceptualization of the case and a crisis plan, - Second phase: to develop strategies both cognitive and behavioral, - Third phase: prevention of relapse.	Determine the effectiveness of the treatment.	Face to face and by telephone.	No
Sáiz et al. (2014) [62]	Treatment. To describe the PSyMAC protocol, a controlled study designed to prevent the recurrence of suicidal behaviors proposed by case management.	All patients older than 18 years to the Emergency Service of a hospital, after having made a suicide attempt.	Suicidal behavior.	Clinical settings.	Over 12 months, ten sessions (one per week).	- Periodical interviews with patients; - Information collection about therapeutic situations; - Reinforce therapeutic adherence; - Coordinate appointments for periodic meetings with the referenced psychiatrist;Encourage the therapeutic return of those patients who have ceased voluntarily their treatment;Facilitate contact with the existing social resources; - Psychoeducation program.	Effectiveness already proven.	Face-to-face group sessions and telephone.	No
Santamarina-Perez et al. (2021) [24]	Prevention. To investigate cognitive differences among adolescents at risk for suicide versus healthy controls and identify cognitive changes associated with response to psychotherapy among adolescents at high risk for suicide.	35 adolescents (12–17 years old) at high risk for suicide, and 14 healthy control adolescents.	Visual memory (lower performance on verbal memory and processing speed may be associated with a high risk for suicide).	Clinical settings, outpatient clinic.	16 weeks.	- Treatment as usual (TAU): routine care (at least one biweekly, 60 min individual sessions) and counseling, elements of cognitive behavioral therapy or psychoeducation; - DBT: individual sessions, skills group-training sessions for adolescents, skills group-training sessions for families, and brief intersession telephone contacts; - TAU + group sessions: individual sessions, group sessions for adolescents, group sessions for families, and intersession telephone contacts.	Concludes that visual memory may be a potential marker of response to treatment in adolescents at high risk for suicide.	Face to face and intersession telephone contacts. Individual and group sessions. Adolescents and parents attended these GS separately.	No
Santamarina-Pérez et al. (2020) [34]	Prevention. To compare the effectiveness of an adapted form of Dialectical Behavior Therapy for Adolescents (DBT-A) and treatment as usual, plus group sessions (TAU + GS) to reduce suicidal risk for adolescents.	Hospital patients aged between 12 and 17 years 11 months with a high risk of suicide and with at least one parent or guardian willing to participate in family sessions.	Frequency of non-suicidal self-injury and number of suicide attempts, changes in the level of functionality, suicidal ideation, and depressive symptoms.	At the hospital.	4 months.	Adapted form of dialectical behavior therapy for adolescents (DBT-A): individual sessions; weekly sessions of group skills training; weekly consultation team meetings for therapists; and telephone consultation service.	Trial design.	Face to face and by telephone.	No

**Table 2 healthcare-12-00792-t002:** Main characteristics of the interventions in the Japanese population.

Study	Main Objective	Target Population	Aspects on Which It Intervenes	Place of Action	Duration	Components	Phase (Proven Effectiveness)	Modality	COVID Inclusion
Furuno et al. (2018) [50]	Treatment. To evaluate whether assertive case-management intervention can reduce the number of repeat episodes of overall self-harm during the whole study period.	914 patients, aged 20 years and older who were admitted to the emergency department.	Episodes of overall self-harm.	Clinical settings.	18 months.	The ACTION-J intervention. Case management consisted of assessment, planning, encouragement, and coordination.	Similar effects on suicide-attempting patients with comorbid Axis I and II psychiatric diagnoses to those among patients who attempted suicide with only an Axis I diagnosis.	Face to face or by telephone.	No
Fujisawa et al. (2013) [46]	Prevention. To evaluate the self-assessed competence and confidence of medical residents about the management of potentially suicidal patients.	114 medical residents, mean age 27.8.	Suicide-intervention skills.	Clinical settings.	2 h.	A brief suicide-management education program for medical residents: lecture session (60 min) about depression and suicide, the five-step principles of mental health first aid: (1) Assess risk of suicide or harm; (2) Listen non-judgmentally; (3) Give reassurance and information; (4)Strengthen the demand for professional assistance."; and (5) Encourage self-help strategies. Role-play session (60 min), discussion, and final Q and A time.	Highlighting the need for improved suicide-management programs for junior medical residents in Japanese hospitals.	Group intervention.	No
Harada et al. (2019) [26]	Prevention. To examine whether the effects of a suicide-prevention education program for junior high school students were moderated by the risk level of students.	28 high-risk minor students and 167 low-risk students.	Suicide risk.	Highschool.	four sessions.	GRIP short version contents: Session 1: Mind Pocket. Coping skills. Session 2: KINO (emotional expression game “KINO”). Session 3: Scenario Contest using the DVD teaching materials. Session 4: Scenario Contest 2, learn how to respond when you notice your friend self-harming.	Quantitative effectiveness of the program verified.	Group intervention.	No
Hashimoto et al. (2021) [35]	Training. To compare the effect of the GTK program to a general mental health lecture that lacked role play and to examine its generalizability.	81 teachers from the Hokkaido University Sapporo Campus (mean age 47.2).	Competence and confidence in managing suicide intervention and behavioral intention as a gatekeeper.	At university.	One session with a 30 min lecture about mental health and 2 h role play.	Psychoeducation about mental health (depression and suicide), didactic lecture on basic gatekeeping skills, a video, role playing, and discussion groups.	Effectiveness already proven in middle and high schools; extend these findings to the university settings.	In person.	No
Hashimoto et al. (2016) [63]	Training. To investigate the effectiveness of the gatekeeper-training program for administrative staff in Japanese universities.	76 administrative staff of Hokkaido University, both sexes, mean age: 36.3	Competence and confidence in the management of suicidal students and behavioral intentions as a gatekeeper.	At university.	2.5 h.	Psychoeducation about depression and suicide, lecture about suicide-intervention skills, a video, and a role-play session.	First study evaluating this program; significant improvement in scores on competence and confidence in the management of suicidal students.	In person.	No
Inui-Yukawa et al. (2021) [27]	Treatment. To examine the effectiveness of assertive case-management intervention in preventing suicidal behavior in self-poisoning patients.	297 patients in the intervention group and 295 in the control group, 20 yearsor older.	Non-suicidal self-harm episodes and suicide attempts.	Clinical settings.	18 months.	Assertive continuous case management. Main contents: planning regular interviews, collecting information on the background and treatment status of each patient and an assessment, offering encouragement to seek psychiatric treatment and the provision of psychoeducation, coordinating appointments with psychiatrists and primary care physicians.	Effective intervention when promptly introduced in a hospital setting following a suicide attempt, and especially effective for self-poisoning patients.	Face to face or by telephone.	No
Kawanishi et al. (2014) [38]	Treatment. To investigate whether assertive case management can reduce reattempts of suicide in people with mental health problems who had attempted suicide and were admitted to emergency departments.	914 adult participants aged 20 years and older who had attempted suicide.	Suicide relapse.	At hospital emergency department.	18 months.	Assertive case management: - Regular contact with participants; - Updates on treatment status and social issues; - Encouragement for adherence to psychiatric treatment; - Coordination of appointments with psychiatrists and primary care physicians; - Encouragement for participants who discontinued psychiatric treatment to return; - Referrals to social services and support organizations, with coordination for resource utilization; - Psychoeducation and information on social resources via a website, including resources for family members.	Assertive case management is feasible only in real-world clinical settings.	Face to face or by telephone.	No
Kawashima et al. (2022) [49]	Training. To investigate the effectiveness of brief online gatekeeper training for Japanese university students.	49 university students (25 in a training group and 24 in a control group), age 21.32.	Knowledge about prevention of suicide, intervention skills, self-confidence, and prevention actions.	At university.	2 months.	Web-based questionnaire that included variables such as suicide-prevention knowledge, skills, self-confidence, and demographic information. Information sheet describing the research objectives following a lecture. Follow-up 2 weeks after the training.	The training group showed an effect in terms of basic knowledge about suicide prevention and self-confidence.	Web-based.	No
Kawashima et al. (2020) [29]	Training. To evaluate the effect of an assertive-case-management training program.	274 medical personnel, mean age 38.09.	Attitudes to suicide prevention, gatekeeper self-efficacy, suicide-intervention skills, and attitudes toward suicide.	At hospital.	2 days (16 h).	Lectures, group workshops, and role-play practice sessions. Identification of risk factors for suicide, communication with suicide attempters, case management, discussion, psychoeducation for suicide attempters, case management in follow-up intervention, psychological state of bereaved family members and others, interprofessional collaboration and self-care in suicide prevention.	Decreased recurrence of suicidal behavior in attempters and improved attitudes toward suicide prevention. Increased self-efficacy and intervention skills after training.	Group training program	No
Nakagami et al. (2018) [64]	Training. To evaluate a newly developed suicide-intervention program among medical staff.	74 medical staff members (42 nurses, mean age 38.74, 20 residents, mean age 26.35, 12 physicians, mean age 36.58).	Improve the detection and referral of at-risk individuals.	At hospitals.	2 h.	Changes in knowledge, perceived skills, and confidence in the early intervention of depression and suicide-prevention.	Significant effects on improving perceived skills and confidence.	In person.	No
Norimoto et al. (2020) [48]	Treatment. To evaluate whether assertive case management can reduce the repetition of suicidal behaviors in patients who had attempted suicide with comorbid Axis I and II diagnoses.	914 participants aged 20 years and older.	Suicidal relapse.	Clinical settings.	18 months.	Assessment, planning, encouragement, and coordination involve the following: (1) Periodic face-to-face or telephone contact during and after emergency-department stays; (2) Gathering information on treatment status and social issues; (3) Encouraging treatment adherence; (4) Coordinating appointments with psychiatrists and primary care physicians; (5) Encouraging treatment return; (6) Referring to social services and support organizations, coordinating resource use; (7) Providing psychoeducation and information on social resources.	Effective and feasible. Reduced incident rate of repeat self-harm.	Internet-based system and face to face or telephone.	No
Nozawa et al. (2022) [25]	Training. To describe a research protocol to investigate the effect of a newly developed internet-delivered online peer GKT program to improve post-secondary student self-efficacy as gatekeepers for suicide countermeasures in Japan.	320 students, 18–29 years old (intervention and control groups).	Self-efficacy as a gatekeeper, literacy of suicide, sense of coherence, stigma, help-seeking styles, psychological distress, self-esteem, resilience, behavior as gatekeepers.	Online.	Six sections (each 85 min).	Contents of the gatekeeper program: - Psychology of people who are mentally ill; - Basic knowledge about depression; - Statistics of suicide; - Social isolation; - The importance of consulting; - Warning signs of suicide; - Risk factors for suicide; - How to call out; - How to listen; - Role play (demo video); - Concepts of referral to care; - Case study; - Available resources.	Results not yet published.	Online intervention.	Yes
Ogawa et al. (2022) [43]	Prevention. To examine the impact of a training initiative aimed at fostering support-seeking behavior among students.	188 students (14 years old). The program involves senior volunteers reading picture books to students, with coordination among local government staff.	Worries and the seeking of social support.	High school.	50 min lesson.	Activity/Content Part 1: Introduction: Explanation of health center and lesson content; Part 2: Stress coping/SOS output lecture and questions about stress awareness, stress coping, and how to seek help; Part 3: Picture-book reading; Part 4: Conclusion. Distribute support-center leaflets. Review of the lesson and write down thoughts.	There was an impact on the awareness (self-disclosure).	Group intervention.	No
Ono et al. (2013) [36]	Prevention. To examine the effectiveness of a community-based multimodal intervention for suicide prevention in rural areas with high suicide rates, compared with a parallel prevention-as-usual control group.	631,133 persons (under 25, 25–65, or over 65 years old).	Reinforce human relationship and connectedness in the community by focusing on building social support networks and health-related resources.	Local governments and local health authorities collaborated and implemented the intervention programs.	3.5 years.	- Leadership involvement: Mayor’s messages, regional suicide-prevention committee, formalized service roles, and promotion of social support networks; - Education and awareness: programs to reduce stigma, enhance suicide-risk recognition, and facilitate help-seeking through public campaigns (events, posters, websites, placards, leaflets, brochures, and lectures); - Gatekeeper training; - Support for high-risk individuals: home visits, regional social gatherings, screening for at-risk individuals, directing them to treatment, and supporting self-help activities for high-risk groups.	Unclear effects on the overall rate ratio of the composite outcome in rural areas where the suicide rate was high.	Community-based.	No
Oyama and Sakashita (2017) [28]	Prevention. To test if a 4-year community-based intervention, including universal depression screening in target areas, and subsequent care and support for those identified as suffering from depression, would reduce suicide rates among middle-aged adults in rural areas with a high suicide rate.	90,000 individuals from different Japanese municipalities’ middle-aged adult population (age range of 40–64 years).	Depression and suicide risk.	Clinical settings.	3 years.	Standardized work plan: distribute public information leaflets and newsletters. Initial screening: depression scale administration. Telephone interview on major depressive episodes. Written feedback via mail. Health professionals contact those with depressive episodes, offering referrals to psychiatrists and support for ongoing treatment.	Statistically greater decrease in suicide rate in the intervention area than the comparison areas. Probably successful in reducing suicide rates.	Screening.	No
Oyama and Sakashita (2016) [65]	Prevention. To explore the long-term impact of a universal screening intervention for depression on suicide rates among older community-dwelling adults.	Japanese adults aged 60 years and older. A total of 41,337 people for intervention and 49,073 for control.	Suicide risk.	Clinical settings.	2-year intervention.	Screening (self-administered questionnaire and telephone interview), educational components (90 min workshops taught by municipal public health nurses and open to the general public, and through local public newsletters), and usual care (regular check-ups).	Long-lasting effects in reducing suicide rates.	Screening.	No
Saigo et al. (2018) [66]	Prevention. To clarify how dysfunctional cognitions associated with depressive symptoms improved over 1 year because of G-CBT.	42 students older than 22 years old.	Depression and dysfunctional cognitions.	At university.	Six sessions (60 min., except for the first and final sessions, which were each 90 min).	Psychoeducation on the CBT theory of the relationship between negative automatic thoughts and psychological symptoms, progressive muscle-relaxation training, cognitive-restructuring training, explanation of attribution theory and training in causal attribution therapy, assertiveness training to improve their social skills.	This G-CBT intervention led to decreases in depression scores.	Group therapy.	No
Sakamoto et al. (2014) [51]	Prevention. To report findings on the effect of a psychoeducational video as a suicide-prevention measure in a Japanese rural town.	2000 residents aged between 30 and 79 years.	Knowledge about local suicide-prevention measures, advice on suicidal ideation and financial issues, attitudes toward suicide, actual and desired familiarity levels with relatives and neighbors, social support, and depressive symptoms.	Residencial.	4 weeks.	Psychoeducational video.	Effectiveness of suicide-prevention measures.	Individual, in-person visits.	No
Shiraga et al. (2013) [67]	Training. To examine the work and mental health of a life-support advisor, and to discuss their role in suicide prevention in the affected area.	19 respondents from local governments.	Physical symptoms, anxiety, and insomnia, social activity, disorders, and depressive tendency.	At a local government.	Started in 2011. Not specified.	The research method was a questionnaire survey (suicide-prevention questionnaire forms), completed after the training session.	The involvement of lifestyle-support counsellors increased.	Individual survey.	No
Sueki and Ito (2015) [68]	Training. To examine the feasibility and effects of online gatekeeping.	139 consultation service users, mean age 23.8 years old.	To promote help seeking in those using web search services.	Online.	Between July and December 2013. Not specified.	Using Google AdWords, keyword-targeted advertisements for a website, and using suicide-related keywords. The advertisements were linked to the website, encouraging the use of an e-mail consultation service. An e-mail address for consultations and phrases to encourage viewers to use such services.	Using suicide- related search advertisements can allow us to contact suicidal Internet users.	Email-based.	No
Sueki et al. (2022) [69]	Prevention. To measure the suicidal ideation and depressive/anxiety tendencies of users of an email-based consultation service for suicide prevention.	167 participants between 10 s and 50 s.	Suicidal ideation.	Online.	4 weeks.	- Advertisement for free expert consultation during specified searches; - Web page outlining email-based consultation services; - Online assessment questionnaire survey. Interacting with service users: Identifying the target, initial approaches, assessing risks and protective factors, and providing support. Detailed gatekeeping procedures and reply-email instructions in an online manual.	Suicidal ideation among users at high risk of suicide decreased approximately four weeks after using the service.	Email-based and phone.	No
Suzuki et al. (2014) [37]	Training. To evaluate the effectiveness of a brief suicide-management training program for Japanese medical residents compared with the usual lecture on suicidality.	114 residents (intervention group n = 65, control group n = 49), mean age 27.5.	Suicidal behavior.	Clinical settings.	2 h.	Structured educational program on managing people with depression and suicidal thoughts. The first part of the program was in lecture format, with factual information on depression and suicide in Japan. Action plans for the management of people with depression and suicidal thoughts. In the second part, a clinical scenario (DVD), role-play, was used. An interactive discussion.	Evaluated the effectiveness of the gatekeeper-training program.	Group intervention.	No
Tachibana et al. (2020) [39]	Prevention. To test the effectiveness of the intervention program proposed to reduce suicidal ideation and improve maternal mental health.	464 women from Nagano’s city, mean age: 31.98 years old.	Prevent suicide in postnatal women at risk of psychosocial problems, reducing suicidal ideation and improving their mental health.	Clinical settings.	During 28 or 60 days postpartum.	Home visits, intervention (psychoeducation, collaboration with family members to create support, commitment to not getting hurt, and support in problem solving, risks, and benefits about psychotropics), and follow-up.	Suggests effectiveness for reducing suicidal ideation and improving maternal mental health.	In person.	No

**Table 3 healthcare-12-00792-t003:** Exhaustive analysis of the outcomes of effectiveness-proven interventions.

	Study	Evaluation Methods	Outcomes	Notable Aspects
	Angora et al. (2022) [30]	Assessment not included in the data-collection protocol for this research.	Percentage reduction and delay of suicide attempts.	Reinforcement of common treatment, providing flexible service tailored to the individual circumstances.
	Cebrià et al. (2013) [56]	Through an interview and considering days elapsed between the first suicide attempt and the percentages.	Time elapsed between initial suicide attempt and subsequent and short-term effects on suicide rates.	Reinforcement of the in-person intervention with the telephone is essential.
	Gabilondo et al. (2020) [22]	Average time until the first suicide reattempt, percentages of patients, survival analysis, and the use of emergency departments’ records.	Increased adhesion to treatment.	Important role of brief contact techniques, like by telephone.
	Jiménez-Sola et al. (2019) [52]	Hospital data records and time measurements.	Decrease in suicide attempts and increase in adherence to treatment.	Effectiveness assessment on a population level, considering broader environmental and contextual factors that may influence outcomes.
	Marco et al. (2022) [40]	Psychological variables’ scales and questionnaires, quality of life index, and subjects’ opinions.	Favors family acceptance and prevented suicidal behaviors.	Positive impact of changes in family members on the state of patients with suicidal behavior.
Spanish interventions	Martínez-Alés et al. (2021) [60]	Observational methods.	Greater therapeutic contact not only improved patient outcomes, but also reduced costs.	Low cost and flexibility of the intervention.
	Martínez-Alés et al. (2019) [44]	Observational methods.	Lower risk of suicide relapse.	Suicide risk prevention through enhanced contact post-discharge from healthcare settings (regular phone calls, home visits, assessments of mental health status, and ensuring connection to appropriate care resources).
	Reijas et al. (2013) [33]	Cohort comparison.	Lower risk of suicidal relapse.	Specific, simple, and economical intervention that involves comprehensive strategies.
	Sáiz et al. (2014) [62]	Psychological scales and numerical data related to suicidal behavior.	Decreased recurrence of suicidal behavior	The inclusion of a psychoeducation group (integrative psychoeducational model) using already-known tools.
	Santamarina-Perez et al. (2021) [24]	Sociodemographic scale, DSM-IV-TR diagnosis, global-level functioning scale, intelligence and verbal memory, and learning tests.	Detection of a risk factor for suicide in adolescents.	Effective results but limited to the study design.
	Harada et al. (2019) [26]	In-person interviews, Japanese version of the BIS/BAS scale, items related to counselling behavior, observation.	Acquisition of emotional and communication abilities.	Inclusion of emotional education and consultation skills in educational environments where the suicidal risk may still be low.
	Hashimoto et al. (2021) [35]	Suicide-intervention-response inventory.	Increase in skills and self-perception of effectiveness in its implementation.	A program focused on teachers rather than students.
	Hashimoto et al. (2016) [63]	Self-administered questionnaire.	Increased capacity to manage cases of suicidal behavior among students.	Compensates for the lack of closeness in university environments.
	Inui-Yukawa et al. (2021) [27]	In-person and telephone interviews.	Reduced incidence and prevalence of suicide.	It emphasizes early intervention, involving collaboration with community resources.
	Kawashima et al. (2022) [49]	Paper-and-pencil psychological assessment measures.	Impact on attitudes towards suicide, skills, and self-efficacy in the application of measures for its prevention.	Inclusion of assertive case management within formal training.
	Kawashima et al. (2020) [29]	True-and-false survey, psychological scales, and inventories.	Improved intervention skills and self-confidence in suicide prevention.	Enrichment of the virtual environment (case studies and role-playing) to allow a gatekeeper prevention program at a distance.
	Nakagami et al. (2018) [64]	Ten yes-or-no questions to evaluate knowledge, six questions on a five-point Likert scale for confidence, and original questionnaires for skills.	Increasement of mental health knowledge, confidence, and skills to prevent suicide.	Considerable effects in a short time intervention focusing on depression and suicidal behavior.
Japanese interventions	Norimoto et al. (2020) [48]	Incidence proportion of the first episode of recurrent suicidal behavior.	Assertive case management was significantly effective for the Axis I group.	Delivering psychoeducational content and information about social resources before.
	Oyama and Sakashita (2017) [28]	Observation of suicide rates.	Probably lower suicide rates.	Detection of cases of depression with a predisposition for suicidal behavior.
	Oyama and Sakashita (2016) [65]	Difference in changes in suicide rates and number of deaths from national registry data.	Long-lasting effects reducing suicide rates.	Importance of depression screening for suicide prevention.
	Saigo et al. (2018) [66]	Paper-and-pencil psychological-assessment measures.	Reduction levels of anxiety and depression that predispose people to suicidal behavior.	Prioritizing cognitive behavioral techniques
	Sakamoto et al. (2014) [51]	Questionnaire prepared by the authors.	Increase in knowledge about suicide.	Brief exclusive use of digital media (video) as prevention measure.
	Shiraga et al. (2013) [67]	Japanese version of the General Health Questionnaire.	Increased protective factors against suicide, specifically social support.	Community involvement in problems that affect peers.
	Sueki and Ito (2015) [68]	Suicidal ideation assessed through text in e-mails.	Increased use of help resources.	A passive procedure from professionals through new technologies to achieve adherence to treatment.
	Sueki et al. (2022) [69]	Online questionnaire survey for sociodemographic data, Japanese self-administered Suicidal Ideation Scale and K6 scale.	Significant reduction in suicidal thoughts and tendencies towards depression/anxiety.	Online counselling interventions more appropriate according to women and more effective in the ideation than in the intention phase.
	Tachibana et al. (2020) [39]	EPDS screening instrument.	Reduced suicide ideation.	Suicide case management for maternal patients encompasses tailored care and support, involving thorough assessment, monitoring, and intervention to address perinatal challenges and mental health issues.

## Data Availability

Not applicable.

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
