# Peer review of "Suicide Interventions in Spain and Japan: A Comparative Systematic Review"

_healthcare, 2024, doi:10.3390/healthcare12070792_

Round 1
Reviewer 1 Report
Comments and Suggestions for Authors
I have comprehensive review of your paper. I have provided constructive feedback aimed at enhancing the quality and impact of your article. I encourage you to consider these suggestions to improve clarity and overall effectiveness. Upon implementing revisions, I am eager to review the updated version. Your commitment to refining your work is admirable, and I am eager to witness the evolution of your article. Best regards.
Abstract
1. The abstract is generally clear and concise. However, consider condensing some sentences to improve readability. For example, in the first sentence, you can combine the information about rising suicide rates in Spain and the aim of the review more succinctly.
2. Provide more details on the search strategy and inclusion criteria. For instance, specify the time frame for the publications (2013-2023), the number of databases searched, and any specific criteria for the inclusion of articles.
3. Since assertive case management was only highlighted in Japan, consider providing a brief explanation or context about what this entails. This would help readers understand the significance of this finding.
4. Consider adding a sentence or two about the implications of the findings for practitioners and researchers in the field. Additionally, mention any recommendations for future research based on the identified gaps or trends.
Introduction
- Consider reorganizing the introduction to enhance the flow. For example, you can start with a general overview of suicide rates in Spain and then introduce the situation in Japan, followed by the need for psychological interventions.
- It would be beneficial to include numerical data consistently throughout the introduction to reinforce the magnitude of the issue. For instance, you've provided specific numbers for suicide rates in Spain, but you might want to add more quantitative information about Japan.
- Provide a concise explanation of what is meant by "psychological interventions" at the beginning of the introduction. Define or give examples of these interventions to ensure clarity for readers who may not be familiar with the term.
- While you've presented the research questions (RQs) at the end, consider integrating them earlier in the introduction to give readers a roadmap of what to expect.
- Conclude the introduction with a clear statement or summary that sets the stage for the systematic review. Clearly state the significance of comparing interventions in Spain and Japan and what potential contributions this research might make to the field.
Methods
1. Provide a brief rationale for selecting the databases you chose (PsycInfo, Web of Science, and Scopus) in terms of their relevance to the research question. This can enhance the transparency of your methodology.
2. Clearly describe the search strategy, including the combination of search terms, Boolean operators, and any filters used. This will help readers understand how you identified relevant studies.
3. Specify the exact search dates for each database to provide a clear timeframe for your literature review. This is important for transparency and reproducibility.
4. Explain why Mendeley Desktop was chosen for the search and screening process. Mention any specific features or advantages that influenced this choice.
5. Provide details on the inclusion/exclusion criteria used during the title and abstract screening. Clarify the criteria that were considered for inclusion or exclusion at this stage.
6. Elaborate more on the nature of the two disputed studies and the criteria used to resolve the disputes. This adds transparency to the decision-making process.
7. Provide a bit more detail on how the risk of bias was assessed for the included studies. Mention if any established tools or criteria were used, and explain how disagreements were resolved.
8. Explicitly state if there were any restrictions on study design (e.g., randomized controlled trials, observational studies) during the selection process.
9. Offer a brief justification for choosing the period between 2013 and 2023. Why was this specific time frame chosen, and how does it contribute to the study's objectives?
10. Clarify whether any software or tools were used for the content analysis, and provide a brief overview of how Braun and Clarke’s six steps were applied.
11. Provide more details on how the qualitative characteristics of interventions and their results were synthesized. What themes and subthemes emerged, and how were they identified?
Discussion
· Consider breaking down the lengthy discussion into subsections to enhance readability. For example, you could have sections dedicated to intervention types, evaluation methods, effectiveness, and considerations for cultural adaptation.
· When discussing the number of retrieved articles and the final number included, consider providing a flowchart or a table to visually represent the screening process. This can help readers better understand the selection criteria and the reasons for excluding certain articles.
· Clearly highlight the key similarities and differences between the Japanese and Spanish interventions. Use tables or bullet points to present comparative information on intervention types, evaluation methods, and outcomes.
· Emphasize the strengths of the methodological diversity in the included studies, such as quasi-experimental designs and observational studies. However, also acknowledge the limitations introduced by these differences, and discuss how they might impact the comparability of outcomes.
· Provide more detail on the feasibility of interventions and the retention of participants. Elaborate on how stable contexts, such as clinical settings or educational institutions, may have contributed to the feasibility of these interventions.
· Explore the implications of the interventions developed after the COVID-19 pandemic in Spain and Japan. Discuss whether there were unique challenges or opportunities presented by the pandemic and how these interventions addressed them.
· Provide more insights into how social support was integrated into interventions. Discuss whether involving relatives or facilitating contact with them had specific positive effects on the outcomes.
· Elaborate on the limitations related to the exclusion of public documents from central and local governments and grey literature. Discuss the potential impact of this exclusion on the comprehensiveness of the review.
· Expand on the suggestions for future research. For example, discuss specific areas where more research is needed, potential methodologies, and the implications for policy development.
· Strengthen your concluding remarks by summarizing the key findings and emphasizing their implications for suicide prevention policies in both countries.
· Ensure that all citations in the text are properly formatted and cross-checked with the reference list.
Reviewer 2 Report
Comments and Suggestions for Authors
Subject: Manuscript Review for "Suicide Interventions in Spain and Japan: A Comparative Systematic Review"
Comments and suggestions for improvement.
In abstract
I recommend the authors clearly articulate specific objectives or research questions guiding the systematic review. Providing more explicit details, such as assessing intervention effectiveness or
The authors should provide more details on the search strategy, including the rationale behind selecting keywords and employing Boolean operators. Addressing any limitations or considerations in the choice of search terms would contribute to the methodological transparency.
Main manuscript
The introduction could benefit from better organization and cohesion. Integrating subsections like "Epidemiology" and "Background and current situation" may improve readability.
Statistical Data Presentation
While the introduction presents statistical data on suicide rates, I recommend the inclusion of more recent data (up to 2023) and a clearer presentation, potentially using tables or graphs for better visualization.
In-depth Justification for Temporal Scope
Provide an in-depth rationale for the chosen timeframe (2013-2023) to enhance reader understanding.
Language Clarity
Some sentences are complex, and simplifying sentence structures would enhance clarity and ensure easy comprehension.
Methods
Expanded Risk of Bias Assessment
The abstract mentions a low risk of bias, but the authors should provide more details on the criteria and process used for assessment. Expanding on this aspect would strengthen methodological rigor.
While the research questions are clear, adding specificity to each question could further guide the research. For instance, detailing the types of interventions and specific prevention elements of interest in each question.
Results
The results section could benefit from a clearer presentation, potentially using tables or figures for better summarization and comprehension. A more detailed analysis of the effectiveness and outcomes of interventions would strengthen the paper. Are there specific interventions that stand out as more effective?
Discussion
The manuscript briefly mentions the impact of COVID-19 on interventions. An in-depth exploration of external factors influencing effectiveness, beyond the pandemic, would add depth to the discussion.
The conclusion could be refined to succinctly summarize the main findings and emphasize key takeaways for policymakers and researchers.
While the paper touches on cultural differences, a more explicit exploration of how cultural factors may influence intervention effectiveness could enhance the discussion.
The brief mention of technology in interventions could be expanded. A more detailed exploration of challenges and opportunities associated with technology-based interventions would be valuable.
Comments on the Quality of English Language
Language Clarity
Some sentences are complex, and simplifying sentence structures would enhance clarity and ensure easy comprehension.
Reviewer 3 Report
Comments and Suggestions for Authors
I read the proposed manuscript with interest. The authors present a systematic literature review aimed at performing a comparison between Spain and Japan in terms of strategies for approaching the problem of suicide. I agree that suicide is a phenomenon of global concern that deserves in-depth studies for the implementation of the best prevention strategies.
The paper is interesting but I would invite the authors to detail better why Japan is the country of reference. The authors say the reason is the reduction in the number of suicides in Japan, but does that mean that in all other European countries, none could be compared? In any case, considering Japan is an agreeable choice. Still, I suggest the authors in the discussions further discuss the socio-cultural differences that exist between Spain and Japan as they may affect suicide and the effect of treatment and prevention measures.
Also, to make the paper even more of a global interest, the authors could better detail what lessons all other countries can learn from this comparison.
